# Oxalic acid blocked the binding of spike protein from SARS-CoV-2 Delta (B.1.617.2) and Omicron (B.1.1.529) variants to human angiotensin-converting enzymes 2

Meng Wang[1,2☯], Huimin Yan[2,3☯], Lu Chen[1,2,3☯], Yu Wang[1,2,3☯], Lin Li[1,2,3☯], Han Zhang[1,2,3☯]*, Lin Miao[1,2,3☯]*

1 State Key Laboratory of Component-based Chinese Medicine, Tianjin University of Traditional Chinese Medicine, Tianjin, China, 2 Institute of Traditional Chinese Medicine, Tianjin University of Traditional Chinese Medicine, Tianjin, China, 3 Key Laboratory of Pharmacology of Traditional Chinese Medical Formulae, Ministry of Education, Tianjin University of Traditional Chinese Medicine, Tianjin, China

☯ These authors contributed equally to this work.
* mmmlin@tjutcm.edu.cn (LM); zhanghan0023@126.com (HZ)

**Data Availability Statement:** All relevant data are within the paper and its Supporting Information files.

## Abstract

An epidemic of Corona Virus Disease 2019 (COVID-19) caused by severe acute respiratory syndrome coronavirus 2 (SARS-CoV-2) is spreading worldwide. Moreover, the emergence of SARS-CoV-2 variants of concern, such as Delta and Omicron, has seriously challenged the application of current therapeutics including vaccination and drugs. Relying on interaction of spike protein with receptor angiotensin-converting enzymes 2 (ACE2), SARS-CoV-2 successfully invades to the host cells, which indicates a strategy that identification of small-molecular compounds to block the entry is of great significance for COVID-19 prevention. Our study evaluated the potential efficacy of natural compound oxalic acid (OA) as an inhibitory agent against SARS-CoV-2 invasion, particular on the interaction of the receptor binding domain (RBD) of Delta and Omicron variants to ACE2. By employing a competitive binding assay *in vitro*, OA significantly blocked the binding of RBDs from Delta B.1.617.2 and Omicron B.1.1.529 to ACE2, but has no effect on the wide-type SARS-CoV-2 strain. Furthermore, OA inhibited the entries of Delta and Omicron pseudovirus into ACE2 high expressing-HEK293T cells. By surface plasmon resonance (SPR) assay, the direct bindings of OA to RBD and ACE2 were analyzed and OA had both affinities with RBDs of B.1.617.2 and B.1.1.529 and with ACE2. Molecular docking predicted the binding sites on the RBD-ACE2 complex and it showed similar binding abilities to both complex of variant Delta or Omicron RBD and ACE2. In conclusion, we provided a promising novel small-molecule compound OA as an antiviral candidate by blocking the cellular entries of SARS-CoV-2 variants.

## Introduction

The pandemic of coronavirus disease 2019 (COVID-19) caused by severe acute respiratory disease coronavirus 2 (SARS-CoV-2) has confirmed cases over 500 million with more than 6

**Funding:** "This work was supported by The National Key Research and Development Project of China (No.2021YFC1712904), National Natural Science Foundation of China (No.82074105) and Research cooperation Project of Guangzhou Xin Huangpu Joint Innovation institute of Chinese Medicine (No.2022OTH009)" The funders had no role in study design, data collection and analysis, decision to publish, or preparation of the manuscript.

**Competing interests:** The authors have declared that no competing interests exist.

million deaths globally by 20 May 2022 [1]. Furthermore, emergence of mutations in SARS-CoV-2 resulted in unexpected disturbs in extremely high infectivity and unexpected ability of immune escape [2–4]. Within distinct variants of concern (VOCs), Delta variant (B.1.617.2) was first reported in India in October 2020 and compared to the wild-type, studies have shown that the Delta variant has nearly 97% increase in the effective reproductive number of virus and thus has more viral loads and higher risks of hospitalization, intensive care unit admission and mortality in infected patients [5, 6]. Omicron variant (B.1.1.529) was first identified in Botswana in November 2021. Although evidence showed that Omicron coronavirus causes milder symptoms than previous variants, it still well-concerns that due its high transmissibility, viral infectivity and immune evasion [7, 8]. Omicron may be ten times more infectious than the original SARS-CoV-2, and twice as infectious as the Delta variant due to its mutations [9]. Studies have also revealed that the Omicron variant is partially resistant to the neutralizing activity of therapeutic antibodies and convalescent sera [10–13], which poses significant threat for the clinical effectiveness of the current vaccines and therapeutic antibodies.

SARS-CoV-2 belongs to the β-coronaviruses, which are enveloped, single, and positive-stranded RNA viruses [14]. Its genome DNA encodes the replicase polyprotein and a series of its structural proteins including Spike (S), Envelope, Membrane, and Nucleocapsid proteins [15]. Among these structural proteins, S protein plays the most critical role during viral attachment, fusion and entry into target cells [16]. It consists of two subunit S1 and S2, and within S1 a receptor-binding domain (RBD) is considered as the direct binding site to the host receptor angiotensin-converting enzyme 2 (ACE2), which will be subsequently proteolytically activated by human proteases and finally help virus entry into the cells [17, 18]. Although a series of options including anti-viral drugs, vaccines, and monoclonal antibodies have been marketed against SARS-CoV-2 [19], the emergence of viral escape mutants continually calls for the development of drugs and vaccines against variants [20]. Targeting the interaction of S protein with ACE2 receptor has thus been considered as a promising target for drug identification that may contribute to the therapeutic strategy of COVID-19 [21].

Natural products are a good source of molecules with low-toxicity and stable activity for therapeutic development [22, 23]. Oxalic acid (OA) is a strong dicarboxylic acid (Fig 1) occurring in various plants and vegetables such as sugar beet and spinach. It has been identified as a

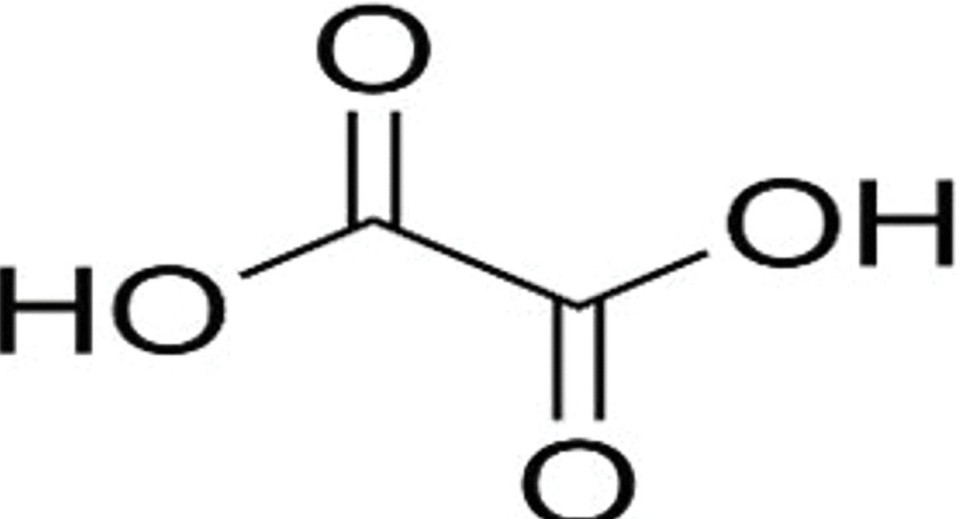

**Fig 1. Chemical structure of oxalic acid.**

lipase inhibitor to reduce the absorption of fat by intestinal cells [24]. However, its anti-viral effect has not been reported. A prospective cohort study published in 2021 revealed COVID-19 patients had enriched OA levels in serum when compared to SARS-CoV-2 negative but with COVID-19 like symptom patients, indicating a potential role of OA in SARS-CoV-2 infection [25]. In this study, the potential inhibitory effect of OA on prevention of SARS-CoV-2 entries was investigated by targeting at the complex of S protein and ACE2 *in vitro* and in pseudovirus assays. The variant specificities of OA among wild type, Delta B.1.617.2 and Omicron B.1.1529 were also compared, its direct binding to either S or ACE2 were determined and the potential binding sites were further predicted as well. Our results will provide evidence whether OA blocked the binding of SARS-CoV-2 invasion, indicating its potential role in drug discovery against COVID-19.

## Materials and methods

### Oxalic acid

Oxalic acid was purchased from YuanYe Bio-Technonlogy Co. Shanghai, China with a purity of $\geq$98% (Cat. No. B20132) and dissolved in dimethyl sulfoxide (DMSO) and stored at-20˚C.

### Cell culture

ACE2 high expressing-HEK293T cells (ACE2$^h$ cells) were kindly provided by Prof Guo Xiaohuan from Tsinghua University. ACE2$^h$ cells were kept in Dulbecco's Modification of Eagle's Medium (DMEM) (Kibbutz Beit Haemek, Israel, Cat. No. 2110050) containing 10% FBS (Invigentech, CA, US, Cat. No. A6907FBS-500) and 1% penicillin-streptomycin (VivaCell, Shanghai, China, Cat. No.C3421-0100), and cultured at 37˚C containing 5% of $CO_2$.

### Cytotoxicity assay

For cytotoxicity assay, ACE2$^h$ cells were seeded into 96-well plates at a density of $1\times10^4$ cells per well and then treated with different concentrations of OA (0.01, 0.1, 1, 10, 100 μM) for 24 h, then 10 μL of Cell Counting Kit (IV08-500, Invigentech, USA) solution was added to each well followed by 2 h of incubation. The relative cell viability was assessed by the detection of the absorbance at 450 nm using a microplate reader (Infinite F50, Thermo Fisher, China). The survival rate of ACE2$^h$ cells was calculated as the following formula: Survival rate = [(OD $_{Treated}$ −OD $_{Blank}$) / (OD $_{Control}$−OD $_{Blank}$)] × 100%.

### Enzyme-linked immunosorbent assay

According to the instruction (ACROBiosystems, Cat. No. EP111/EP-115, Beijing, China), various concentrations of OA were added to the wells in an ACE2 pre-coated plate followed by addition of HRP-SARS-CoV-2 Spike RBD. After incubation at 37˚C for 1h, wells were washed and the substrate was then added. After incubation at 37˚C for 20 min, the reaction was finally terminated by the addition of stop solution. The absorbance at 450 nm was measured immediately by the microtiter plate reader (Infinite F50, Thermo Fisher, China). The neutralizing antibody provided by ACROBiosystems was used as the positive control. The inhibition rate was calculated as the following formula: Inhibition rate = (1-OD Sample/ OD Negative Control) × 100%.

### Pseudovirus entry assay

Pseudovirus from SARS-CoV-2 B.1.617.2 (Cat. No. FNV3718) and B.1.1529 (Cat. No. FNV4122) were purchased from FUBIO (Jiangsu, China). Pseudovirus (2×107 TFU/ml) were pre-incubated with different concentration of OA or ACE2-Fc (GenScript, Cat. No. Z03516,

Nanjing, China) as a positive control in a 96-well plate for 1 h at room temperature. ACE2$^h$ cells were then seeded at $3 \times 10^4$/well into the plate with the pseudovirus-OA mixture and incubated for additional 6 h. Then the medium was replaced with fresh medium and incubated for another 48 h. The GFP images were captured by BioTek Lionheart FX Automated Microscope (Agilent Technologies, USA). Cells were harvested and the luciferase activity was measured using the luciferase reagent (Promega, Shanghai, China, Cat. No. G7941) by the microtiter plate reader (PerkinElmer, Victor XLight). The inhibition rate was calculated as the following formula: Inhibition rate = (1- Signal of positive control—Blank control/Signal of Negative control—Blank control) × 100%.

## Surface plasmon resonance

Surface plasmon resonance (SPR) measurement was performed at 25˚C using a BIAcore T200 instrument. RBDs of Delta B.1.617.2 (Sino Biological, Beijing, China, Cat. No.40592-V08H90, 50 μg/ml) and Omicron B.1.1.529 (GenScrip, Nanjing, China, Cat. No. Z03728, 50 μg/ml) and ACE2 (Sino Biological Inc., China. Cat. No. 10108-H05H, 102 μg/ml) protein were immobilized on a CM5 sensor chip (Cytiva, Uppsaia, Sweden, Cat. No. 29104988) respectively, and a blank channel was employed as a negative control for each assay. Different concentrations of OA were then injected into the immobilized biosensor chips. The contact time was 120s and dissociation time was 180s. The 1:1 binding model was used to assess the binding kinetics. KD values were calculated with a kinetics model by BIAcore T200 analysis software.

## Molecular docking

**Ligand preparation.** 3D structure of OA was obtained and downloaded from the Chemical library ChemSpider (http://www.chemspider.com/) and converted to PDB files. "Prepare ligands" module in Discovery Studio (DS) 2020 was applied with the candidates followed by "minimize ligands" module.

**Protein preparation.** High-resolution X-ray crystal structures of S-ACE2 complexes (PDB ID: 7V8B and 7T9L) were taken as targets. To optimize the structure, the complex was initially processed by "prepare protein" module in DS 2020, followed by supplementation of the missing sidechains, removal of the water molecules and protonation suitable at pH 7.4. According to the previous findings [26–30], the active binding sites of Delta RBD-ACE2 complexes and Omicron RBD-ACE2 complexes were set in Tables 1 and 2 separately.

## Docking

The molecular docking of OA was performed at the interface of ACE2-RBD of Delta (7V8B) and Omicron (7T9L) through the semi-flexible docking by CDOCKER module in DS 2020. In the Parameters panel, the "Input Receptor" was set to 7V8B or 7T9L. Parameter "Input Ligands" was set to Molecule: All. The "Top Hits" parameter was expanded and "Pose Cluster Radius" was set to 0.5. Then the different conformations of OA were docked one by one by scanning the entire protein surface to get the ligands binding specifically to the interface of ACE2-RBD of Delta (7V8B) and Omicron (7T9L). The binding interaction energies and their potential binding sites were recorded.

## Statistical analysis

The raw data from the cytotoxicity assay and enzyme-linked immunosorbent assay were analyzed using GraphPad Prism 8.0 software. Log (inhibitor) vs. response——Variables slope (four parameters) was used to fit the inhibition curve. All values are presented as the

**Table 1. The active binding sites of Delta RBD-ACE2 complexes.**

| RBD (Delta) | ACE2 | Category | Types | References |
|---|---|---|---|---|
| THR500 | TYR41 | Hydrogen Bond | Conventional Hydrogen Bond | [26, 27] |
| GLY502 | LYS353 | Hydrogen Bond | Conventional Hydrogen Bond | [26, 27] |
| GLN498 | | | Conventional Hydrogen Bond | [26, 27] |
| GLY496 | | | Conventional Hydrogen Bond | [26, 27] |
| GLY446 | GLN42 | Hydrogen Bond | Conventional Hydrogen Bond | [26, 27] |
| TYR449 | ASP38 | Hydrogen Bond | Conventional Hydrogen Bond | [26, 27] |
| GLN493 | LYS31 | Hydrogen Bond | Conventional Hydrogen Bond | [26, 27] |
| | GLU35 | Hydrogen Bond | | [27] |
| LYS417 | ASP30 | Hydrogen Bond | Salt Bridge | [26, 27] |
| TYR489 | TYR83 | Hydrogen Bond | Conventional Hydrogen Bond | [26, 27] |
| ASN487 | GLN24 | Hydrogen Bond | Conventional Hydrogen Bond | [26, 27] |
| ALA475 | SER19 | Hydrogen Bond | Conventional Hydrogen Bond | [26, 27] |

mean ± the standard (SD). One-way ANOVA was used to determine the statistical significance between different group in the luciferase activity assay and cytotoxicity assay. The numbers of experimental replicates are shown in the figure legends. Sensorgram figures from the SPR assays were generated with the GraphPad Prism 8.0 software.

## Results

### OA inhibited RBDs of SARS-CoV-2 Delta (B.1.617.2) or Omicron (B.1.1529) bound to ACE2

OA was tested in vitro for its inhibitory effect on RBD-ACE2 interaction by ELISA. As Fig 2A shown, OA blocked the interaction between Delta RBD and ACE2 in a dose-dependent manner with an $IC_{50}$ of 3.039 mM (Fig 2A). Similarly, OA also significantly inhibited ACE2 binding to Omicron RBD with an $IC_{50}$ of 1.059 mM (Fig 2B). In contrast, no inhibitory effect of OA was detected on ACE2 binding to wild type of SARS-CoV-2 RBD (S1 Fig in S1 Appendix). Recombinant neutralizing antibody provided within was used as a positive control, and it strongly inhibited the formation of ACE2-RBD complexes (S2 Fig in S1 Appendix).

### OA suppressed the entries of pseudovirus of SARS-CoV-2 Delta (B.1.617.2) and Omicron (B.1.1529) into ACE2h cells

Before pseudovirus infection assay, cell viability was first determined in response to OA and data showed that no significant cytotoxic effect was detected in ACE2$^h$ cells even at 100 μM

**Table 2. The active binding sites of Omicron RBD-ACE2 complexes.**

| RBD (Omicron) | ACE2 | Category | Types | References |
|---|---|---|---|---|
| ARG493 | Glu35 | Hydrogen Bond | Salt Bridge | [28–30] |
| TYR453 | HIS34 | Hydrogen Bond | Conventional Hydrogen Bond | [26] |
| TYR501 | | | | [28] |
| HIS505 | GLY354 | Hydrogen Bond | Conventional Hydrogen Bond | [26] |
| ASN477 | SER19 | | | [26] |
| SER496 | LYS353 | Hydrogen Bond | Conventional Hydrogen Bond | [26, 30] |
| GLY502 | | | | [26, 29] |
| THR500 | TYR41 | Hydrogen Bond | Conventional Hydrogen Bond | [26] |
| ARG498 | GLN42 | Hydrogen Bond | Conventional Hydrogen Bond | [26, 28, 30] |

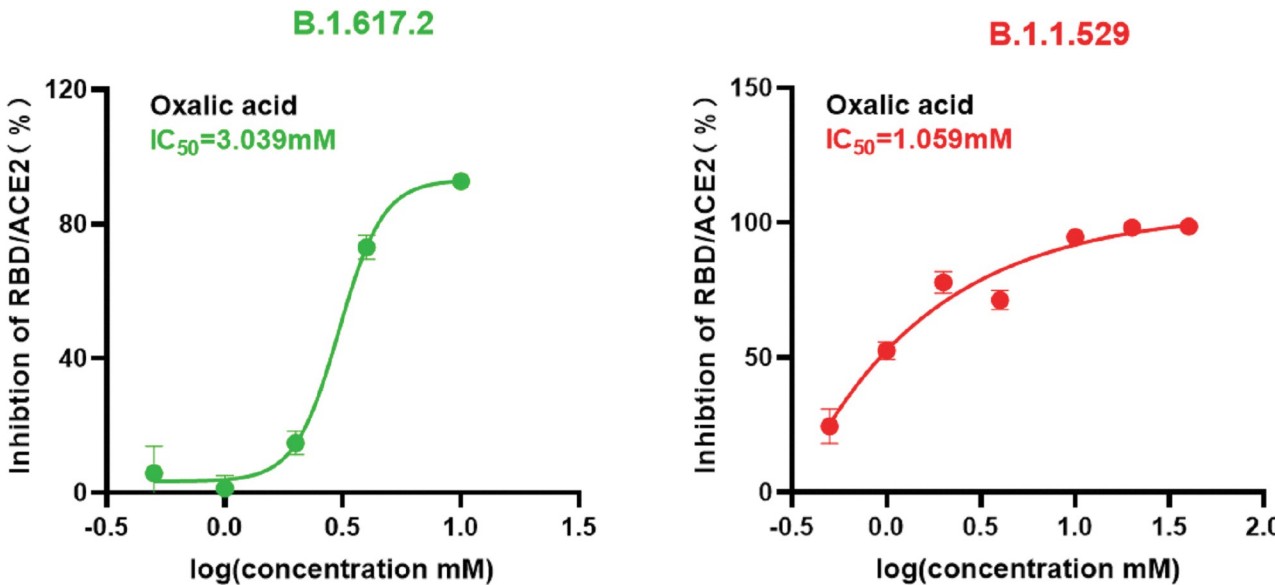

**Fig 2. Effects of OA on the interaction between ACE2 and SARS-CoV-2 Spike RBD from Delta (B.1.617.2) and Omicron (B.1.1529).**
Representative inhibitory curves of ACE2 binding to SARS-CoV-2 RBDs of Delta RBD (A), Omicron RBD (B) in the presence of OA determined by ELISA.

(Fig 3). Pseudovirus infection was then performed to detect the inhibitory effect of OA on the viropexis of SARS-CoV-2 Delta and Omicron variants. As shown in Fig 4, OA inhibited the entry of SARS-CoV-2 Delta pseudovirus into ACE2[h] cells in a concentration-dependent manner with IC$_{50}$ of 26.04 μM. Consistently, the expression of green fluorescent protein in ACE2[h] cells was significantly reduced caused by ACE2-Fc or OA in the images photographed by a fluorescence microscope (Fig 4C).

Similarly, ACE2[h] cells were infected with SARS-CoV-2 pseudovirus Omicron (B.1.1529), and OA significantly reduced the luciferase activity when compared to the control (Fig 5A). Photographed images were shown in Fig 5B under the concentration of 10 μM as well.

## OA directly bound to ACE2 and RBDs of SARS-CoV-2 Delta (B.1.617.2) and Omicron (B.1.1529)

To identify the direct interactions of OA on ACE2 and RBD, we further addressed the SPR assay. OA brought about a concentration-dependent resonance change when flowing through the sensor chip coated with either ACE2 (Fig 6A), RBD Delta (Fig 6B) or Omicron (Fig 6C). The $K_D$ values were then calculated by fitting the affinity data at various concentrations of OA and recorded in Fig 6D.

## The predicted binding sites of OA to the complex of ACE2 with Delta or Omicron RBDs

Recently, it has been well reported that the binding structures of the Delta and Omicron RBDs with ACE2 [20, 21], which provide as important targets for candidate screening and site identification. As shown in Fig 7, the active binding sites were set in Delta RBD-ACE2 (Fig 7A) or Omicron RBD-ACE2 complex (Fig 7B). In details, Delta RBD interacted with ACE2 at the residues THR500, GLY502, GLN498, GLY496, GLY446, TYR449, GLN493, LYS417, TYR489, ASN487 and ALA475 with TYR41, LYS353, GLN42, ASP38, LYS31, GLU35, ASP30, TYR83,

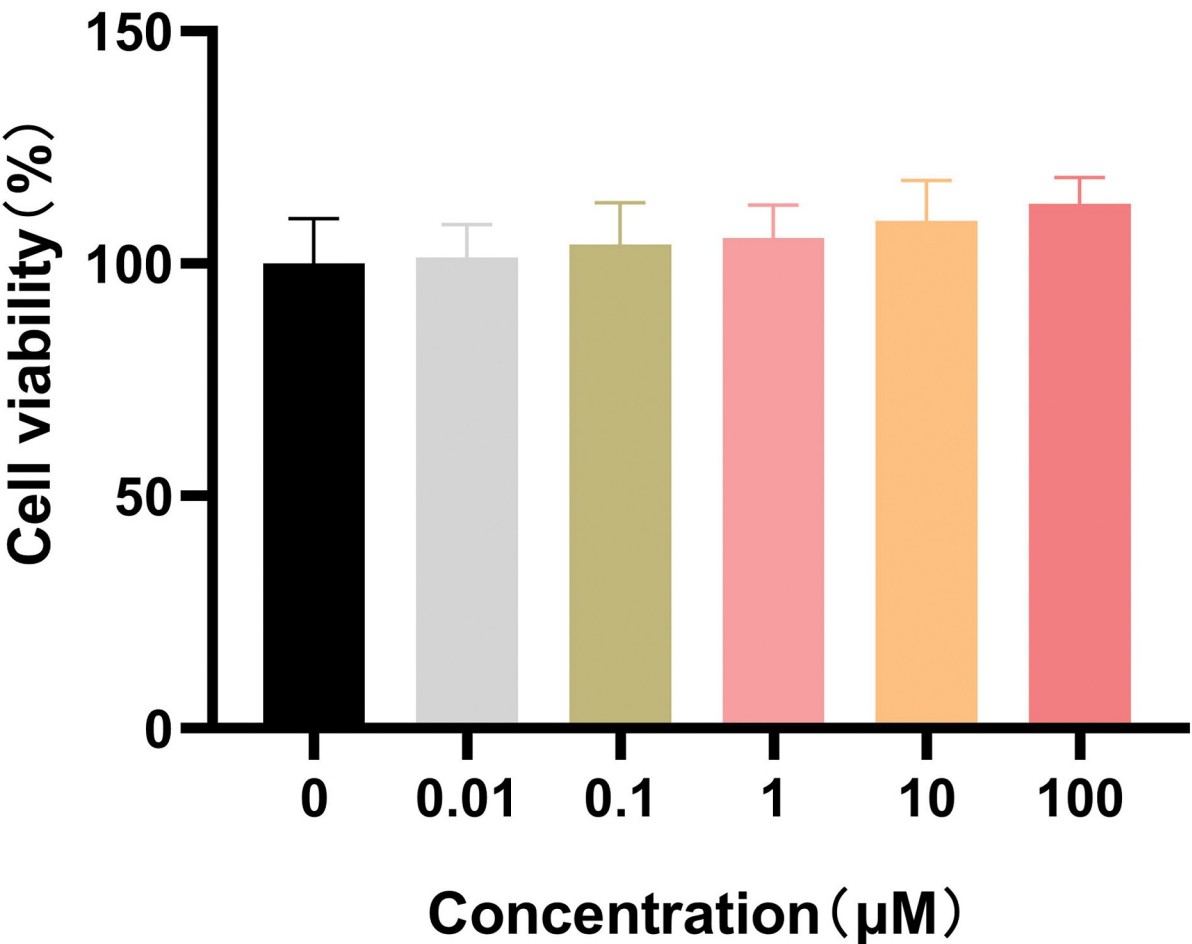

**Fig 3. Effect of OA on ACE2[h] cell viability.** ACE2[h] cells were pretreated with different doses of OA and incubated for 24 h. Viability of ACE2[h] cells was detected at $OD_{450nm}$. The proliferation ration was calculated. The experiments were repeat three times. Data are presented as mean ± *SD*.

GLN24 and SER19 in ACE2 via hydrogen bonds, and the residues ASN501, GLY502, TYR505, PHE456, LEU455, TYR453, and PHE486 stabilized with ARG357, ASN330, ASP355, GLY354, GLU37, THR27, HIS34, PHE28, MET82 of ACE2 through non-bonded interactions. The residues involved in the interaction of Omicron RBD and ACE2 included the ARG493, TYR453, TYR501, SER496, GLY502, ASN477, THR500, ARG498 in RBD and GLU35, HIS34, GLY354, LYS353, SER19, TYR41 and GLN42 in ACE2 via hydrogen bonds, as well as the residues PHE456, LEU455, HIS505, TYR489, ALA475, ASN487, GLY476, TYR449 and PHE486 of Omicron RBD with ASP38, THR27, PHE28, TYR83, ARG357, LYS31, ASP355, GLN24, LEU79, MET82 of ACE2 through non-bonded contacts (Fig 7B).

Next, molecular docking was applied to visually inspect and predict the interfering sites of OA on the complexes of Delta or Omicron RBD with ACE2. As shown in Fig 8A, OA interacted with the residue LYS353 of ACE2 via the conventional hydrogen bond. A carbon hydrogen bond with Delta RBD residues ARG403 and GLY496 and an attractive charge with residues LYS353 and ARG403 were formed with OA as well. The docking scores of different conformation was listed in Table 3 and the average value was calculated to be -30.29 kcal/mol.

In Omicron RBD-ACE2 complexes, OA bound to the residue LYS353 (Fig 8B) via the conventional hydrogen bond. It also formed a carbon hydrogen bond with residues HIS34 and

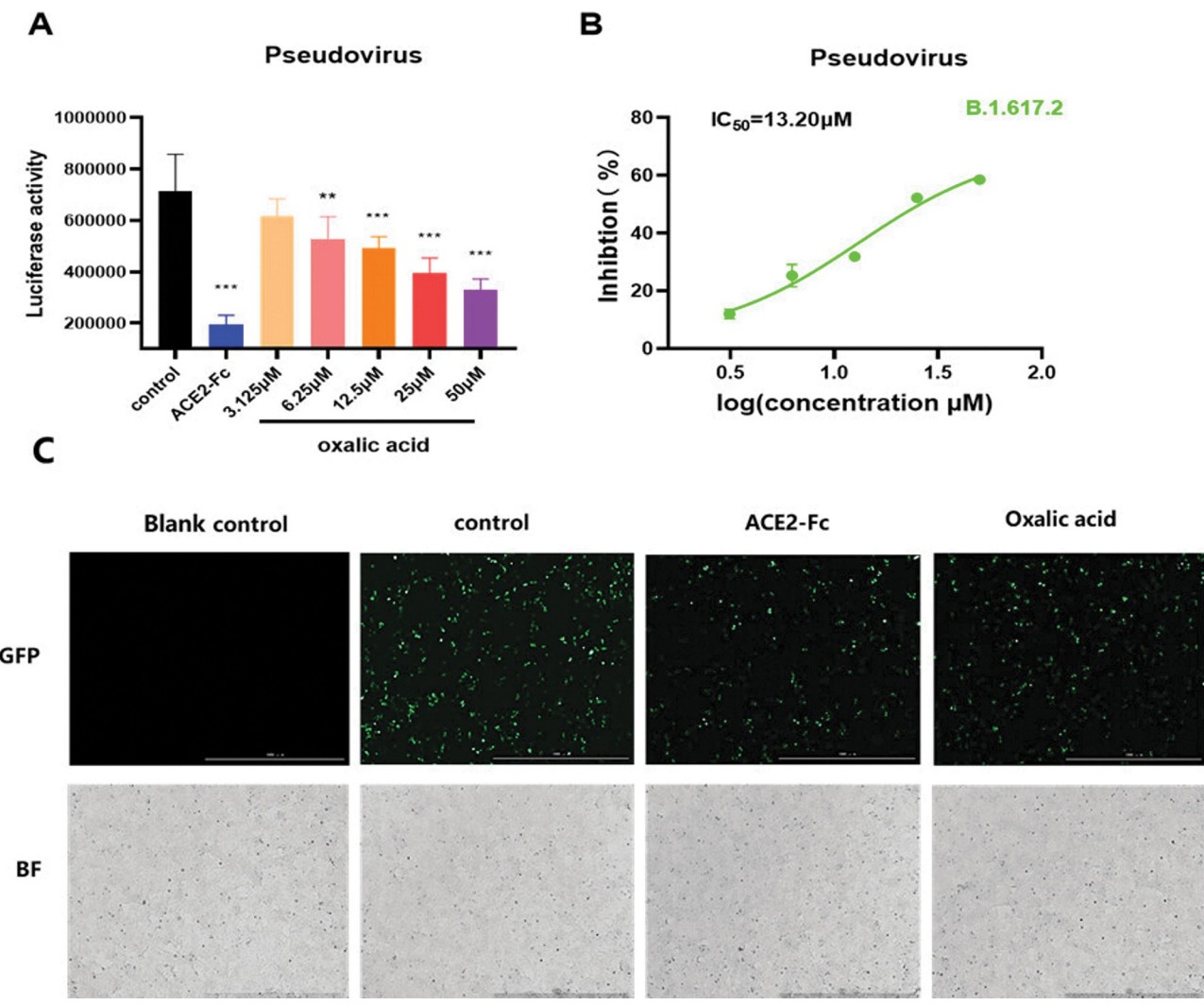

**Fig 4. The effect of OA on the entry of SARS-COV-2 Delta (B.1.617.2) pseudovirus into ACE2^h cells.** (A) The entrance of SARS-CoV-2 Delta pseudovirus into ACE2^h cells was evaluated by the luciferase activity after treated with ACE2-Fc and different concentrations of OA. (B) Inhibition rate was calculated and showed in curves. (C) Representative images were photographed at × 4 magnification using the fluorescence microscope. Data were presented as mean ± S.D. $**p < 0.01$, $***p < 0.001$, compared with control.

ARG403 and an attractive charge with LYS353 and ARG403 residues. The docking scores of different conformations was listed in Table 3. The average value was calculated to be -28.59 kcal/mol. There was a comparable binding affinity of OA interacting with the complex ACE2-RBD Delta and Omicron in this prediction. The average docking scores and the residues interacting with OA were summarized in Table 4.

## Discussion

With the high mutation rate of the virus and the increased risk of immune viral escape, emerging SARS-CoV-2 variants [23, 31] exhibited antiviral resistant and threatened the global health care [32, 33]. Our study first identified a potential role of natural small molecule OA on blocking the binding of Delta and/or Omicron spike to ACE2. Recently, a report found that a traditional drug cysteamine prescribed to decrease intraliposomal cystine accumulation [34]

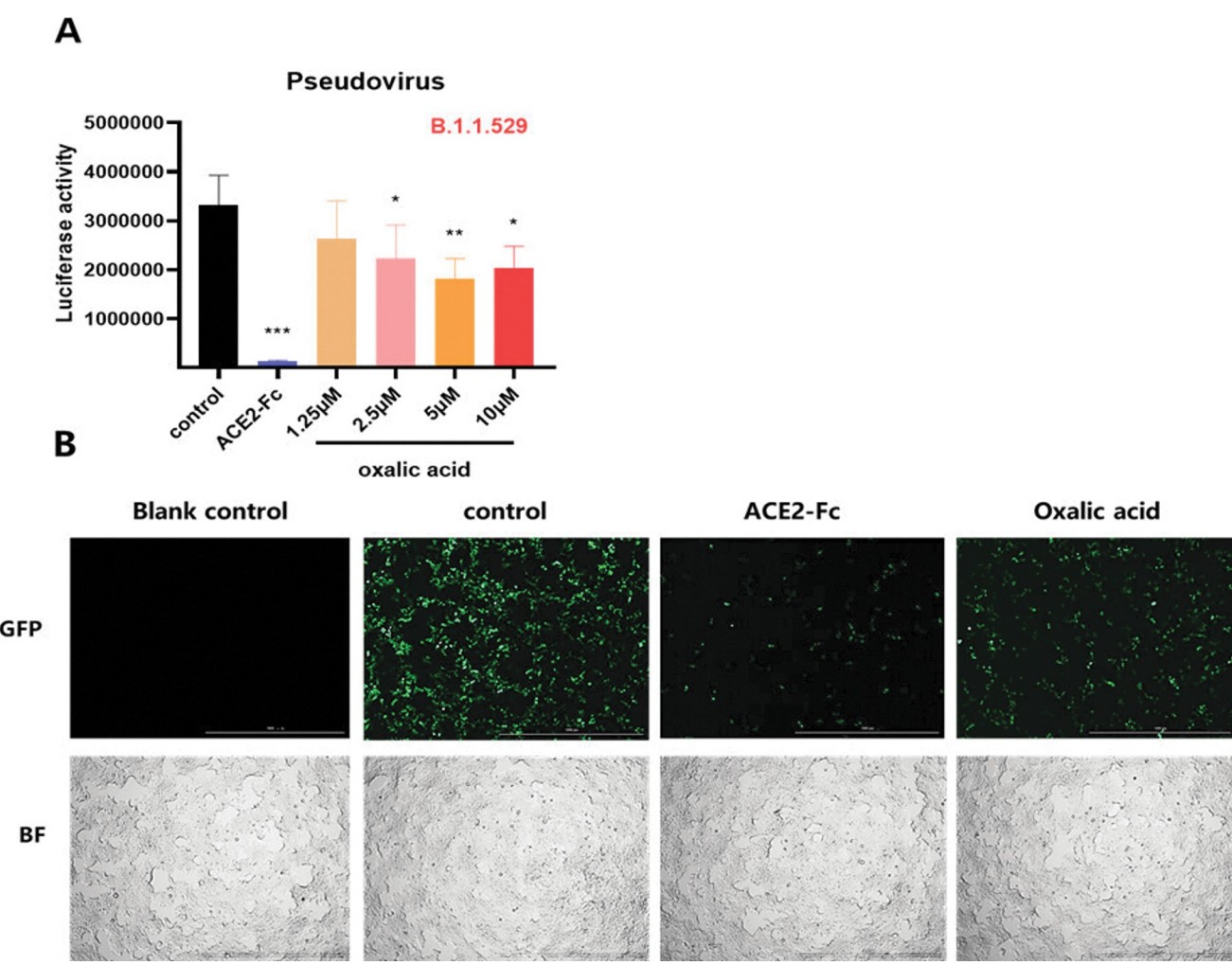

**Fig 5. The effect of OA on the entry of SARS-COV-2 Omicron (B.1.1.529) pseudovirus into ACE2$^h$ cells.** (A) The entrance of SARS-CoV-2 Delta pseudovirus into ACE2$^h$ cells was evaluated by the luciferase activity after treated with ACE2-Fc and different concentrations of OA. (B) Representative images were photographed at × 4 magnification using the fluorescence microscope. Data are presented as mean ± S.D. *$p <$0.05, **$p < 0.01$, ***$p < 0.001$, compared with control.

exerted direct antiviral actions against SARS-CoV-2 Delta and Omicron variants, in addition to the wild type virus [35]. Using multiple and appropriate screening approaches to extend existed drug applicability as well as identify novel candidate drugs may provide evidence as a future candidate for prevention of SARS-CoV-2 variant infections. However, more cellular and animal assays will be needed to further confirm their inhibitory effects, which are allowed to perform only in a biosafety level-3.

Blocking of the interaction between RBD and ACE2 by inhibitors may be due to the binding to RBD or ACE2, or both. Here, in our study, we first found that OA effectively prevented the binding of the RBD from Delta and omicron variants to the ACE2 receptor by competitive ELISA (Fig 2), indicating the possibilities of OA on interaction with either RBD or ACE2. In functional assay, pseudovirus infection data confirmed that interfering with ACE2-RBD complex by OA contributes to the SARS-CoV-2 viral entrance to cells (Figs 4 and 5), as we expected. However, we are still uncertain on the direct interaction of OA with RBD and ACE2. Based on the SPR assay, we further demonstrated that OA directly binding to both RBD and

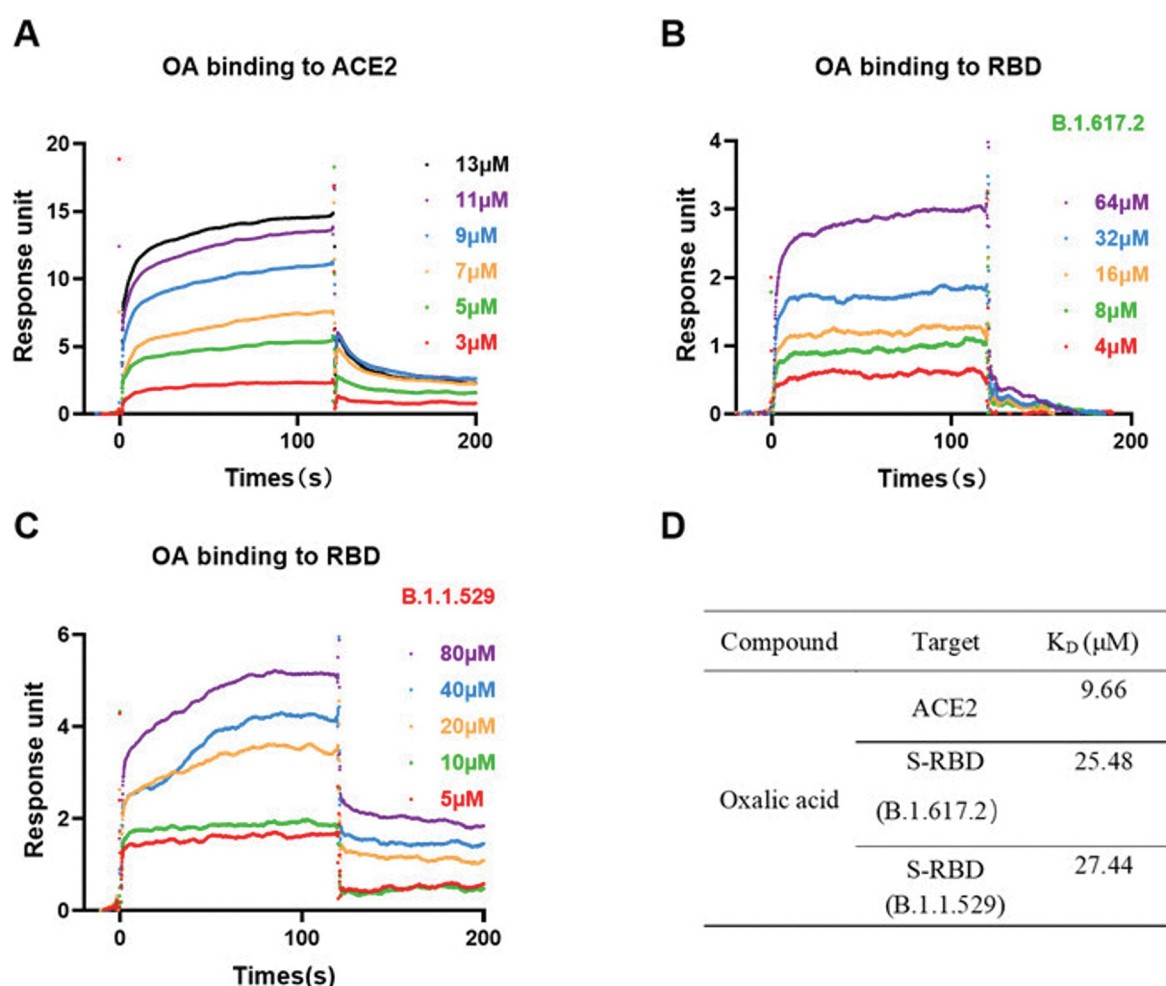

**Fig 6. The binding characters of OA on ACE2 and RBDs of Delta and Omicron.** The SPR titration curves of OA interacted with ACE2 protein (A), Delta RBD protein (B) and Omicron RBD protein (C) were shown. The affinity constants of OA to Delta, Omicron and ACE2 protein were calculated and presented (D).

ACE2 (Fig 6). The $K_D$ values indicated that OA exhibited a higher affinity for ACE2 than those of Delta and Omicron RBDs. It is interesting that the higher affinity of OA to ACE2 than to the Delta and Omicron RBDs, indicating a potential optimized role of OA on blocking the host infection by suppression of the receptor in human. As new variants continue to emerge during the spread of infection, it may be more rational and effective to target the ACE2 receptor.

As far as we have known, ACE2 is part of the renin–angiotensin–aldosterone system (RAAS) involved in regulating cardiovascular processes through reduction of blood pressure [36]. It is expressed and located in the cell membranes in several organs such as the lungs, heart, and kidneys. A series of ACE2 inhibitors have been applied in searching for neutralization of the SARS-CoV-2 virus entries to limit the infection and replication. Here in our study, we found that ACE2 was one of OA's targets (Fig 6A) to interfere with the interaction between the spike protein and ACE2. However, further investigation regarding the effect of OA on the RAAS is required in the future.

We performed the molecule docking assay to predict the binding structures of OA on RBD-ACE2 binding interfaces of Delta and Omicron. The computed binding conformations

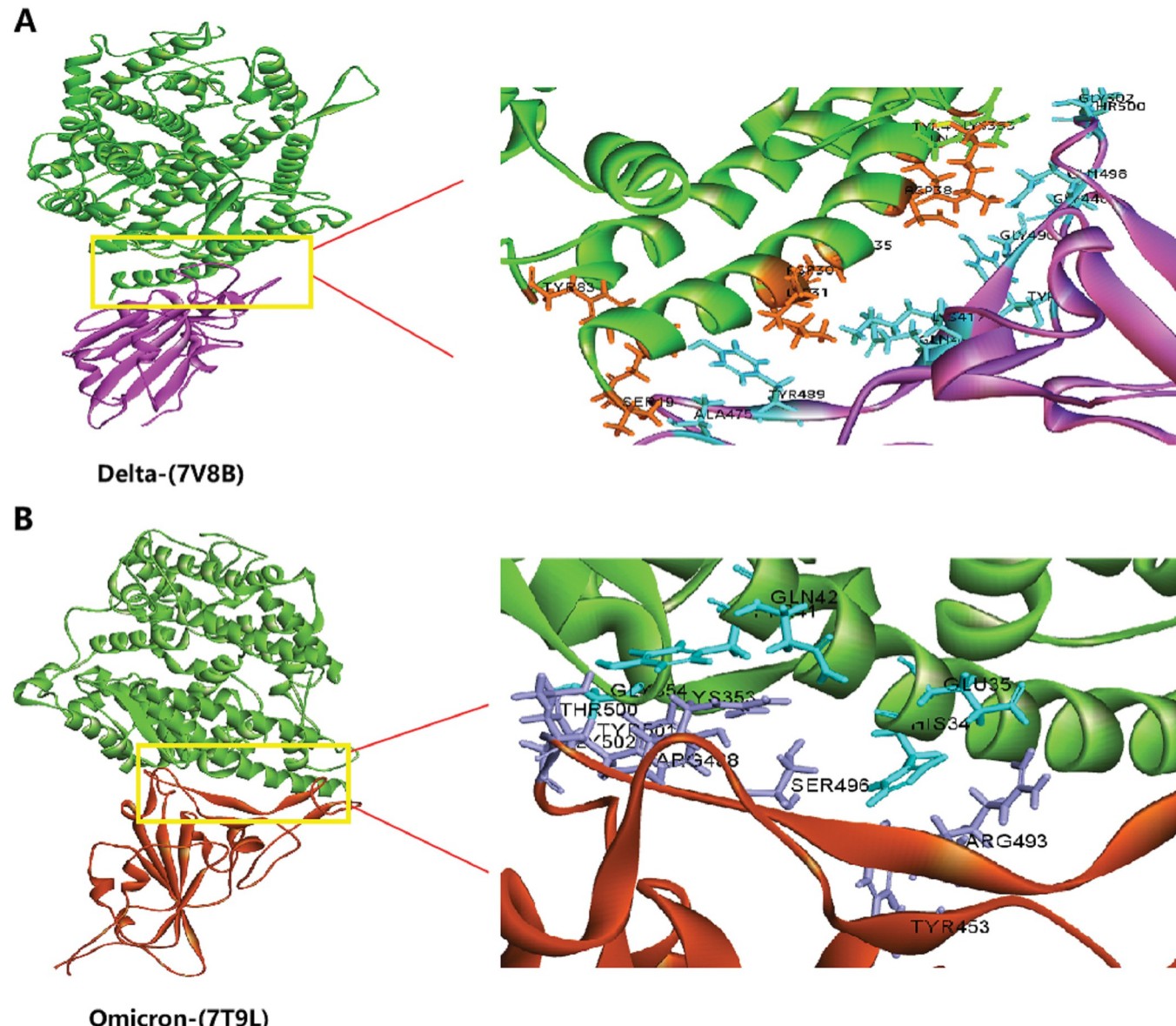

**Fig 7. SARS-CoV-2 Delta or Omicron spike protein in complex with human ACE2.** Delta (A) and Omicron (B) RBD binding to ACE2 interface were marked.

appeared with similar poses (Fig 8), and interaction energy scores were comparable as well (Table 4), indicating an equal binding affinity of OA to the complex ACE2-RBD Delta and ACE2-RBD Omicron. We also presented the amino acid residues in ACE2-RBD complexes involved in the interaction with OA interfering, which may be some clues for the design of neutralizing agents for SARS-CoV-2 and further confirmation assay are needed.

OA is an organic compound found in many plants such as leafy greens, vegetables, fruits, cocoa, nuts, and seeds. It is widely used as an acid rinse in laundries, where it is effective in removing rust and ink stains because it converts most insoluble iron compounds into a soluble complex ion. So far, no anti-viral effect of OA has been reported yet. Here, in our study, we first reported that OA has abilities to directly bind Delta (B.1.617.2) and Omicron (B.1.1.529)

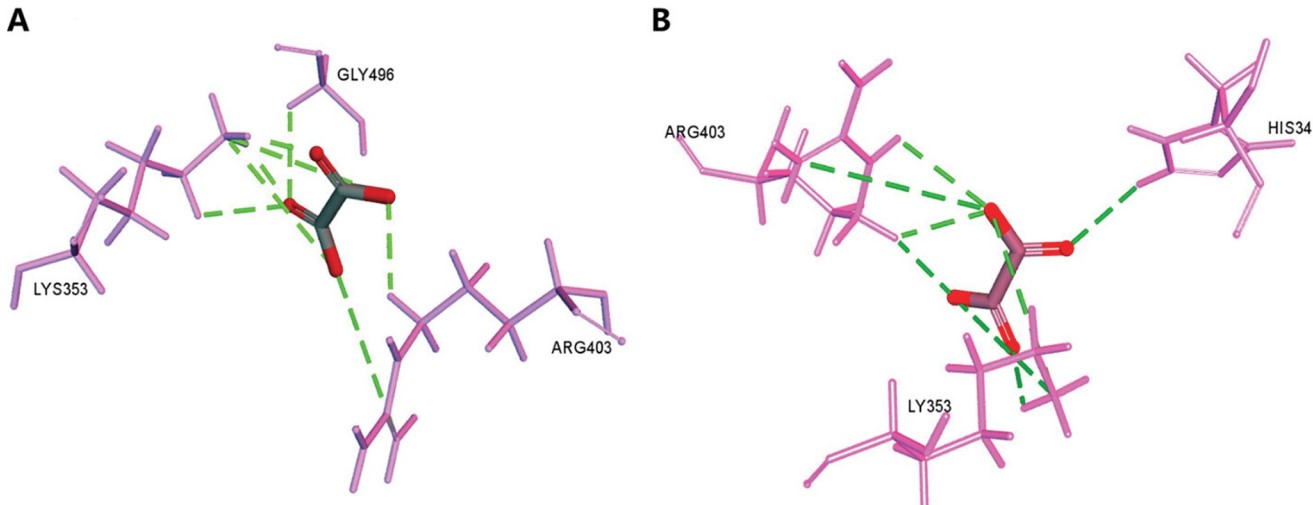

**Fig 8. Molecular docking of OA with RBD-ACE2 binding interfaces.** The predicted binding sites of OA to the complex of ACE2 with Delta (A) or Omicron (B) RBD were shown.

RBD as well as ACE2, by which it effectively blocked SARS-CoV-2 variant entry to cells and may be considered as an anti-SARS-CoV-2 candidate for COVID-19 drug development.

## Conclusion

OA was identified as a novel active constituent which blocked both the binding of SARS-CoV-2 Delta (B.1.617.2) and Omicron (B.1.1.529) RBD to ACE2 *via* residues ARG403, GLY496 and LYS353 in Delta RBD-ACE2 complexes and residues ARG403, HIS34 and LYS353 in Omicron RBD-ACE2 complexes, suggesting the therapeutic candidate of OA for COVID-19 treatment.

**Table 3. The interaction energy of different conformation of OA interacted with RBD-ACE2 complexes of Delta or Omicron.**

| Natural product | Variants | Interaction energy (kcal/mol) |
|---|---|---|
| OA | Delta | -33.9641 |
| | | -31.2595 |
| | | -29.8519 |
| | | -29.9661 |
| | | -29.4299 |
| | | -27.3075 |
| | Omicron | -30.7625 |
| | | -29.7297 |
| | | -29.7642 |
| | | -28.5466 |
| | | -28.4514 |
| | | -28.8786 |
| | | -28.4508 |
| | | -28.2398 |
| | | -26.702 |
| | | -26.3966 |

**Table 4. The predicted binding sites of OA interacted with ACE2 and RBD of Delta or Omicron complexes.**

| | RBD | ACE2 | Distance (Å) | Category | Types | Interaction energy (kcal/mol) |
|---|---|---|---|---|---|---|
| Delta | | LYS353 | 1.899 | Hydrogen Bond | Conventional Hydrogen Bond | -30.29 |
| | | | 4.603 | Electrostatic | Attractive Charge | |
| | ARG403 | | 2.533 | Hydrogen Bond | Carbon Hydrogen Bond | |
| | | | 5.083 | Electrostatic | Attractive Charge | |
| | GLY496 | | 2.812 | Hydrogen Bond | Carbon Hydrogen Bond | |
| Omicron | | HIS34 | 2.992 | Hydrogen Bond | Carbon Hydrogen Bond | -28.59 |
| | | LYS353 | 2.097 | Hydrogen Bond | Conventional Hydrogen Bond | |
| | | | 4.685 | Electrostatic | Attractive Charge | |
| | ARG403 | | 2.924 | Hydrogen Bond | Carbon Hydrogen Bond | |
| | | | 5.237 | Electrostatic | Attractive Charge | |

## Supporting information

**S1 Appendix.**
(DOCX)

## Acknowledgments

The authors would like to acknowledge the support of Tsinghua University for providing the ACE2 high expressing-HEK293T cells.

## Author Contributions

**Formal analysis:** Meng Wang.

**Methodology:** Huimin Yan, Lu Chen.

**Resources:** Yu Wang, Han Zhang, Lin Miao.

**Supervision:** Lin Li.

**Writing – original draft:** Meng Wang.

**Writing – review & editing:** Lin Miao.

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
