## [Decision Letter · Decision Letter 0]

13 Feb 2023

PONE-D-23-01760Oxalic acid blocked the binding of spike protein from SARS-CoV-2 Delta (B.1.617.2) and Omicron (B.1.1.529) variants to human angiotensin-converting enzymes 2PLOS ONE

Dear Dr. Miao,

Thank you for submitting your manuscript to PLOS ONE. After careful consideration, we feel that it has merit but does not fully meet PLOS ONE’s publication criteria as it currently stands. Therefore, we invite you to submit a revised version of the manuscript that addresses the points raised during the review process.

We look forward to receiving your revised manuscript.

Kind regards,

Erman Salih Istifli, PhD

Academic Editor

PLOS ONE

Journal Requirements:

Sub-lineages of the SARS-CoV-2 Omicron variants: Characteristics and prevention - https://doi.org/10.1002/mco2.172

oxalic acid - https://www.britannica.com/science/oxalic-acid

(among others)

In your revision ensure you cite all your sources (including your own works), and quote or rephrase any duplicated text outside the methods section. Further consideration is dependent on these concerns being addressed.

" The funders had no role in study design, data collection and analysis, decision to publish, or preparation of the manuscript."

Additional Editor Comments:

Dear Dr. Lin Miao

Reviewers have now completed their revisions on your paper. You will see that they are advising that you revise your manuscript. If you are prepared to undertake the work required, I would be pleased to reconsider my decision.

Please include continuous line numbers in your manuscript to facilitate editorial handling and reviewing.

Yours sincerely,

Assoc. Prof. Dr. Erman Salih Istifli

Academic Editor

PLOS ONE

Reviewers' comments:

Reviewer's Responses to Questions

**Comments to the Author**

1. Is the manuscript technically sound, and do the data support the conclusions?

Reviewer #1: Yes

Reviewer #2: Partly

2. Has the statistical analysis been performed appropriately and rigorously? 

Reviewer #1: Yes

Reviewer #2: No

3. Have the authors made all data underlying the findings in their manuscript fully available?

Reviewer #1: No

Reviewer #2: No

4. Is the manuscript presented in an intelligible fashion and written in standard English?

Reviewer #1: Yes

Reviewer #2: No

5. Review Comments to the Author

Reviewer #1: The authors describe the very interesting ability of oxalic acid to inhibit the binding of the spike protein from SARS-COV-2 Delta and Omicron-Variant to ACE2 and the therefore the uptake of the virus. As small natural compound, oxalic acid is an interesting antiviral drug candidate for SARS-COV-2.

The performed experiments show nicely demonstrate the effect on binding to ACE2 as well as uptake into cells.

The manuscript is nicely written, nevertheless the phrasing could be partially revisited to further increase the overall readability.

I recommend to publish after minor revision.

1. Table 3 mentioned in line 148 containing the oxalic acid interactions is missing.

2. In the method section for Molecular Docking/Docking line 269, please provide further details on the CDOCKER method and the applied settings, i.e. what means semi-flexible ? ligand? protein? If possible please provide some reference to the method.

3. The amino acid labels in Fig7 are not readable. Please increase the font size or use different representation. For example, single letter amino acid code might be sufficient. If still not suitable, maybe not highlighting all amino acids with name as they are already provided in table 1 and 2 but mostly focus on the highlighting of the position of the oxalate interaction site. That would be alternatively a suggestion for an addon to figure 8 in order to illustrate the relative position of the oxalate binding site. Also, here please increase the label size.

4. I assume the assays were performed as replicates please provide the information.

Reviewer #2: The authors wrote a manuscript on a timely topic and tried to suggest an antiviral natural compound, oxalic acid (OA),

against COVID-19 variants Delta and Omega. The study includes experimental and computational methods, however, the main problem is that the computational studies are not designed well to support the experimental findings. The grammar should also be revised as there are many unclear sentences due to wrong grammar. Another important thing is that Table 3 is missing in the text. The comments and suggestions are included in the attached text. To sum up, I suggest some major revisions for the manuscript to be accepted.

6. PLOS authors have the option to publish the peer review history of their article (what does this mean?). If published, this will include your full peer review and any attached files.

Reviewer #1: No

Reviewer #2: No

---

## [Author Response · Author response to Decision Letter 0]

17 Apr 2023

1. line78：formula could be given

Response: Thank you for reminding of us. The original file was in low format. We have re-uploaded the formula as Fig 1 in the revised manuscript. Please check.

2. line81：the relation of OA and covid-19 could be added as in the paper:

Shi D, Yan R, Lv L, Jiang H, Lu Y, Sheng J, Xie J, Wu W, Xia J, Xu K, Gu S, Chen Y, Huang C, Guo J, Du Y, Li L. The serum metabolome of COVID-19 patients is distinctive and predictive. Metabolism. 2021 May; 118:154739. doi: 10.1016/j.metabol.2021.154739. Epub 2021 Mar 2. PMID: 33662365; PMCID: PMC7920809.

Response: Thank you very much for the literature you provided, which help us find the relationship between COVID-19 and oxalic acid. We have supplemented its findings of OA reported in the paper and labeled them in the introduction part started from Line 81 to 84.

3. Line95：Fig. 2A

Response: Thank you for your reminding. After your suggestion, we have checked the original data and found one of the values in OA in 100 mM group has been mis-inputted before analysis. We have re-corrected it and replaced the statistical analysis. The refreshed figure was presented in Fig 2A in the revised version.

4. Line113：Fig.5 inhibition curve

Response: Thank you for your question. Actually, we performed the pseudovirus luciferase assay with different doses of OA in a range from 1.25 to 80μM. However, the inhibitory curve did not be well-fitted (as below). Therefore, we did not show a curve but the histogram instead. 

5. Line122：Fig. 6D stronger binding to ACE2 should be discussioned

Response: Thank you for the constructive comment. Following your suggestion, we have made corresponding discussion in the Discussion section started from Line 179 to 183.

6. Line127：not clear how the binding site was determined（using which tool）and side chains are missing in the figures.

(1) Response: Thank you for the question. We are sorry not to illustrate clearly in the previous manuscript. Here, we supplemented the setting condition and the reference in the Molecular Docking of Material part. Briefly, the active site definition of Delta RBD to ACE2 was determined based on the following publications Han P et al. Cell 2022 and Celik I et al. Biology 2021 (See the reference list below). We summarized the detail residues involved in Table 1 in the revised version and also presented the reference together in the Table 1 below. Similarly, the binding sites of Omicron RBD to ACE2 was selected based on the following publications Han P et al. Cell 2022, Kumar S et al. Journal of medical virology 2022, Lupala CS et al. Biochemical and biophysical research communications 2022 and Mannar D et al. Science 2022 (See the reference list below). Table 2 in the revised version was shown for the detail residues and we also supplemented the reference together in the Table 2 below.

Reference:

[1] Han P, Li L, Liu S, Wang Q, Zhang D, Xu Z, Han P, Li X, Peng Q, Su C, Huang B, Li D, Zhang R, Tian M, Fu L, Gao Y, Zhao X, Liu K, Qi J, Gao GF, Wang P. Receptor binding and complex structures of human ACE2 to spike RBD from omicron and delta SARS-CoV-2. Cell. 2022 Feb 17;185(4):630-640.e10.

[2] Celik, I., Yadav, R., Duzgun, Z., Albogami, S., El-Shehawi, A. M., Fatimawali, Idroes, R., Tallei, T. E., & Emran, T. B. (2021). Interactions of the Receptor Binding Domain of SARS-CoV-2 Variants with hACE2: Insights from Molecular Docking Analysis and Molecular Dynamic Simulation. Biology, 10(9), 880. 

[3] Kumar, S., Thambiraja, T. S., Karuppanan, K., & Subramaniam, G. (2022). Omicron and Delta variant of SARS-CoV-2: A comparative computational study of spike protein. Journal of medical virology, 94(4), 1641–1649.

[4] Lupala, C. S., Ye, Y., Chen, H., Su, X. D., & Liu, H. (2022). Mutations on RBD of SARS-CoV-2 Omicron variant result in stronger binding to human ACE2 receptor. Biochemical and biophysical research communications, 590, 34–41.

[5] Mannar, D., Saville, J. W., Zhu, X., Srivastava, S. S., Berezuk, A. M., Tuttle, K. S., Marquez, A. C., Sekirov, I., & Subramaniam, S. (2022). SARS-CoV-2 Omicron variant: Antibody evasion and cryo-EM structure of spike protein-ACE2 complex. Science (New York, N.Y.), 375(6582), 760–764.

Table 1 The active binding site of ACE2 and Delta spike protein complexes 

RBD

(Delta) ACE2 references

THR500 TYR41 [1][2]

GLY502 LYS353 [1][2]

GLN498 [1][2]

GLY496 [1][2]

GLY446 GLN42 [1][2]

TYR449 ASP38 [1][2]

GLN493 LYS31

GLU35 [1][2]

 [2]

LYS417 ASP30 [1][2]

TYR489 TYR83 [1][2]

ASN487 GLN24 [1][2]

ALA475 SER19 [1][2]

Table 2 The active cores of ACE2 and Omicron spike protein complexes 

 RBD

(Omicron) ACE2 references

Site ARG493 Glu35 [1][3][4] [5]

 TYR453 HIS34 [1]

 TYR501 [3]

 HIS505 GLY354 [1]

 ASN477 SER19 [1]

 SER496 LYS353 [1][5] 

 GLY502 [1] [4]

 THR500 TYR41 [1]

 ARG498 GLN42 [1] [3] [5]

 

7. Line129、132、137、146、147、148: resides/reside

Response: Thank you very much for discovering these errors. We apologize for these grammatical problems and have corrected them based on your suggestions.

8. Line 145-147,151-152: docking should be repeated and average scores should be given

Response: Thank you for your suggestion. Following your advice, we have attached additional Table 3 to show all the scores getting from molecular docking assay. The average scores for Delta and Omicron with ACE2 were described in the Result part (Line146-147 and Line151-152).

9. Line 149: better to give in kcal/mol units

Response: Thank you for your correction. We have modified the unit.

10. Line 147,152: There is no Table 3 in the text or supp.info

Response: Thank you so much for your reminding. I'm very sorry that Table 3 was lost in the previous version. Here we have supplemented the missing information and numbered in Table 4 in the revised manuscript. Thank you so much for your reminding.

11. Line 156-161: this sentence belongs to introduction.

Response: Thank you for your comment. We have deleted the sentence in the Discussion Part, and the relative message in the sentence was then transferred to the Introduction Part.

12. Line 192: This is not a virtual screening docking but just a molecular docking because only one molecular docking because only one molecule is tested against binding to certain targets

Response: Thank you for your correction. We have revised the description “virtual molecular docking” to “molecular docking”.

13. Line 195: Table 3

Response: I'm very sorry that Table 3 was lost in the previous version. Here we have supplemented the missing information and numbered in Table 4 in the revised manuscript. Thank you so much for your reminding.

14. Line 209-212: grammar should be corrected, computational results not mentioned.

Response: Thank you very much for discovering these errors. We apologize for these grammatical problems. Furthermore, the conclusion was revised with supplementary computational results. The final summarized sentence is “OA was identified as a novel active constituent which blocked both the binding of SARS-CoV-2 Delta (B.1.617.2) and Omicron (B.1.1.529) RBD to ACE2 via residues ARG403, GLY496 and LYS353 in Delta RBD-ACE2 complex and ARG403, HIS34 and LYS353 residues in Omicron RBD-ACE2 complex, suggesting the therapeutic candidate of OA for COVID-19 treatment.”

15. Line 227: No information on how many replicas were used. The error bar is wide in Fig2A Delta variant and needs to be explained and discussed. 

Response: Thank you so much for your remindings. N=3 was supplemented in Line 455 in the revised manuscript. And for the error bar in Fig 2A, according to your suggestion, we have checked the original data and found one of the values in OA in 100 mM group has been mis-inputted before analysis. We have re-corrected it and replaced the statistical analysis. The refreshed figure was presented in Fig 2A in the revised version.

16. Line 261: “and” and “or” are used wrongly through out the text and the meanings are not clear.

Response: Thank you for your correction. We have carefully checked the word “and” and “or” and some mis-using positions have been corrected. Please check.

17. Line 277-278: How? To optimize the structure, the complex was initially processed by prepare protein” module in DS 2020, followed by supplementation of the missing sidechains, removement of the water molecules and protonation suitable at pH 7.4.

Response: Thank you for the question. To optimize the structure, the buttons "clean protein" and "prepare protein" were clicked sequentially on the Discovery Studio software, the software will then automatically supplement missing sidechains of protein, remove the water molecules and protonate suitable in the chain at pH 7.4. It is a DS 2020 internal procedure which immediately response to the request required from a button instruct.

18. Line 281: Needs to be more detailed, for example why the wt spike RBD was not used in docking studied to verify the experiments? Only the variants were used according to the results and methods do not address this issue

(1) Needs to be more detailed, for example why the wt spike RBD was not used in docking studied to verify the experiments?

Response: Thank you for your comments. The details of docking were described started from Line 284 to 289. 

All the related information mentioned here have been supplemented in the Material and method part in Red. Here the information are also shown below： Set the Input Ligands parameter to Molecule: All, and set the other parameters to the default values in“Prepare ligands” module. 

Parameter Name Parameter Value

Input Ligands Molecule: All

Change Ionization True

Generate Tautomers True

Generate Isomers True

Fix Bad Valencies True

Generate Coordinates 3D

Parallel Processing False

(2) Only the variants were used according to the results and methods do not address this issue

Response: Thank you for pointing the question. Yes, the variant Delta and Omicron were included in the assay instead of the wild type due to the low inhibitory activity of OA on WT RBD binding to ACE2 shown in Supplementary Figure 1 (see below as well). Therefore, we predicted the block effect of OA on wild type may not the main targets and all the assays in the manuscript were performed in variants not wild type.

19. Line 290: Not clear, more details needed

Response: Thank you for your suggestions. We have carefully supplemented the details of this section started from Line 291-300.

20. Line 471-473: Mistakes in grammar

Response: Thank you for your correction. We have carefully checked the word “and” and “or” and some mis-using positions have been corrected. Please check.

21. Line 476: Molecular docking? There is no virtual screening

Response: Thank you for your correction. Thank you for your correction. We have revised the description “virtual molecular docking” to “molecular docking”.

22. Figure 7: side chains should be shown

Response: Thank you for your request. The binding structures of RBD-ACE2 complex, particularly the active binding residues in the chains, have been fully shown in the Figure 7. Please check whether it fits.

---

## [Editor Report · Decision Letter 1]

24 Apr 2023

PONE-D-23-01760R1Oxalic acid blocked the binding of spike protein from SARS-CoV-2 Delta (B.1.617.2) and Omicron (B.1.1.529) variants to human angiotensin-converting enzymes 2PLOS ONE

Dear Dr. Lin Miao,

Thank you for submitting your manuscript to PLOS ONE. After careful consideration of the rebuttal letter that you prepare against the first round of revisions, I feel that your manuscript has now been greatly improved. Therefore, I invite you to submit a minor-revised version of the manuscript that addresses the points I raised below.

1. In Tables 3 and 4, please add the minus (-) sign before the Interaction energy (kcal/mol) values. Positive interaction energy values are not acceptable, since they indicate an unfavorable interaction.2. Please rephrase the sentence "When the variants have been keeping emerging during it spread, it is much smart and effective to target on the conservation receptor" between lines 182 - 183, as "As new variants continue to emerge during the spread of infection, it may be more rational and effective to target the ACE2 receptor."

We look forward to receiving your revised manuscript.

Kind regards,

Erman Salih Istifli, PhD

Academic Editor

PLOS ONE
---

## [Author Response · Author response to Decision Letter 1]

25 Apr 2023

1.In Tables 3 and 4, please add the minus (-) sign before the Interaction energy (kcal/mol) values. Positive interaction energy values are not acceptable, since they indicate an unfavorable interaction.

Response: Thank you for your comments. Actually, in both Table 3 and 4, Interaction energy (kcal/mol) values were all below zero, and we also labelled value unit in the corresponding title (-Interaction energy). However, it may be not clear enough and may further confuse the readers. Following your question, we have revised the description of table title from ‘-Interaction energy’ to ‘Interaction energy’, and the values were thus presented with mimus in the following data. Please check.

2.Please rephrase the sentence "When the variants have been keeping emerging during it spread, it is much smart and effective to target on the conservation receptor" between lines 182 - 183, as "As new variants continue to emerge during the spread of infection, it may be more rational and effective to target the ACE2 receptor."

Response: Thank you for the suggestion and help. We have revised the description you mentioned here following your kindly provided. Please check.

---

## [Editor Report · Decision Letter 2]

2 May 2023

Oxalic acid blocked the binding of spike protein from SARS-CoV-2 Delta (B.1.617.2) and Omicron (B.1.1.529) variants to human angiotensin-converting enzymes 2

PONE-D-23-01760R2

Dear Dr. Miao,

We’re pleased to inform you that your manuscript has been judged scientifically suitable for publication and will be formally accepted for publication once it meets all outstanding technical requirements.

Kind regards,

Erman Salih Istifli, PhD

Academic Editor

PLOS ONE

Additional Editor Comments (optional):

After the minor revision, the manuscript by Miao et al. can be published now.
---

## [Editor Report · Acceptance letter]

10 May 2023

PONE-D-23-01760R2 

Oxalic acid blocked the binding of spike protein from SARS-CoV-2 Delta (B.1.617.2) and Omicron (B.1.1.529) variants to human angiotensin-converting enzymes 2 

Dear Dr. Miao:

I'm pleased to inform you that your manuscript has been deemed suitable for publication in PLOS ONE. Congratulations! Your manuscript is now with our production department. 

Kind regards, 

on behalf of

Assoc. Prof. Dr. Erman Salih Istifli 

Academic Editor

PLOS ONE